# Dendritic Cells Pre-Pulsed with Wilms’ Tumor 1 in Optimized Culture for Cancer Vaccination

**DOI:** 10.3390/pharmaceutics12040305

**Published:** 2020-03-28

**Authors:** Terutsugu Koya, Ippei Date, Haruhiko Kawaguchi, Asuka Watanabe, Takuya Sakamoto, Misa Togi, Tomohisa Kato, Kenichi Yoshida, Shunsuke Kojima, Ryu Yanagisawa, Shigeo Koido, Haruo Sugiyama, Shigetaka Shimodaira

**Affiliations:** 1Department of Regenerative Medicine, Kanazawa Medical University, Uchinada, Kahoku 920-0293, Japan; koya@kanazawa-med.ac.jp (T.K.); dat-pey@kanazawa-med.ac.jp (I.D.); harukawa@kanazawa-med.ac.jp (H.K.); asuka-w@kanazawa-med.ac.jp (A.W.); taku0731@kanazawa-med.ac.jp (T.S.); m-togi@kanazawa-med.ac.jp (M.T.); tkato@kanazawa-med.ac.jp (T.K.J.); 2Center for Regenerative medicine, Kanazawa Medical University Hospital, Uchinada, Kahoku 920-0293, Japan; ken1-y@kanazawa-med.ac.jp; 3Center for Advanced Cell Therapy, Shinshu University Hospital, Matsumoto, Nagano 390-8621, Japan; kojishun@shinshu-u.ac.jp (S.K.); ryu@shinshu-u.ac.jp (R.Y.); 4Department of Gastroenterology and Hepatology, The Jikei University School of Medicine, Kashiwa, Chiba 277-8567, Japan; shigeo_koido@jikei.ac.jp; 5Department of Cancer Immunology, Osaka University Graduate School of Medicine, Osaka 565-0871, Japan; sugiyama@sahs.med.osaka-u.ac.jp

**Keywords:** cell-based drug delivery, dendritic cells, tumor-associated antigens, Wilms’ tumor 1

## Abstract

With recent advances in cancer vaccination therapy targeting tumor-associated antigens (TAAs), dendritic cells (DCs) are considered to play a central role as a cell-based drug delivery system in the bioactive immune environment. Ex vivo generation of monocyte-derived DCs has been conventionally applied in adherent manufacturing systems with separate loading of TAAs before clinical use. We developed DCs pre-pulsed with Wilms’ tumor (WT1) peptides in low-adhesion culture maturation (WT1-DCs). Quality tests (viability, phenotype, and functions) of WT1-DCs were performed for process validation, and findings were compared with those for conventional DCs (cDCs). In comparative analyses, WT1-DCs showed an increase in viability and recovery of the DC/monocyte ratio, displaying lower levels of IL-10 (an immune suppressive cytokine) and a similar antigen-presenting ability in an in vitro cytotoxic T lymphocytes (CTLs) assay with cytomegalovirus, despite lower levels of CD80 and PD-L2. A clinical study revealed that WT1-specific CTLs (WT1-CTLs) were detected upon using the WT1-DCs vaccine in patients with cancer. A DC vaccine containing TAAs produced under an optimized manufacturing protocol is a potentially promising cell-based drug delivery system to induce acquired immunity.

## 1. Introduction

Despite significant advances in cancer therapy such as surgical techniques, radiotherapy, and systemic therapy including immune checkpoint inhibitors [1,2,3,4,5,6], it remains extremely challenging to treat advanced cancers involving organ systems and distant metastasis. Therapeutic peptide vaccines targeting tumor-associated antigens (TAAs) for cancer immunotherapy have been in development for decades [7]. The efficacy of peptide vaccines is dependent on the peptide selected for TAAs, peptide formulation, and combined adjuvant [8,9]. Wilms’ tumor 1 (WT1) molecules are expressed in various types of solid tumors; therefore, a peptide vaccination targeting this molecule has priority as an immunotherapy for cancer patients [10]. Hailemichael et al. reported that incomplete Freund’s adjuvant, Montanide ISA51, in cancer peptide vaccines induced persisting vaccine depots [11]. Montanide ISA51 was used in peptide formulation [9], triggers specific T-cell sequestration, dysfunction, and deletion at the vaccination site. In other words, peptide vaccination may be insufficient to recruit TAAs specific cytotoxic T lymphocytes (CTLs) to tumor sites, appropriate adjuvants and/or delivery systems may be useful to exert antitumor immunity.

On the basis of cancer vaccination as an immunotherapy, dendritic cells (DCs) play central roles in antigen uptake, processing, and delivery to primed naïve T cells in lymphoid organs [12]. DCs are native adjuvants to immunogenicity and act as cell-based drug delivery systems in cancer immunotherapy. Antigen-presenting-cell-based immunotherapy with active DCs has been reported for the induction of efficient immunity against cancer antigens [13]. DC vaccines using tumor lysates are expected to uptake TAAs [14,15,16], and it has been reported that autologous DCs pulsed with oxidized autologous whole-tumor cell lysate amplify T cells against the individual neoantigens of patients [17]. DC vaccines pulsed with WT1 peptides, synthesized artificially, were previously shown to be safe and feasible with few adverse reactions in patients with advanced cancer [18,19,20,21,22,23,24,25]. Furthermore, WT1-pulsed DC vaccines primed with low-dose rhG-CSF are expected to achieve higher acquired immunogenicity [25]. The establishment of a standardized protocol for the production of DC vaccines targeting TAAs is considered useful for the development of DC vaccines equipped with antitumor immunity.

The large-scale preparation of a DC vaccine for clinical use with homogeneous, mature, and functional profiles is a prerequisite for achieving efficacious cancer immunotherapy [26]. Autologous monocyte-derived DCs are conventionally manufactured using granulocyte-macrophage colony-stimulating factor (GM-CSF) and interleukin (IL)-4 and matured via exposure to OK-432 (streptococcus preparation; pharmaceutical agent) and prostaglandin E2 (PGE2) using an adherent culture standardized protocol for clinical trials [18,19,20,21,22,23,24]. Mechanistically, OK-432 promotes the functional maturation of immature IL-4-DCs through ligation of Toll-like receptor (TLR) 2, TLR4 [27], and TLR9 [28], and this maturation is correlated with the upregulated expression of CD80, CD83, and CD86 [29,30,31], thereby promoting the effective induction of antigen-specific T cells [29]. The combined treatment of mature IL-4 DCs with OK-432 and PGE2 results in the upregulation of CD197 (CCR7), which is associated with migration to lymph nodes [31]. OK-432 also induces the production of IL-12 by matured DCs without increasing the production of immunosuppressive cytokines such as IL-10.

Floating non-adherent cells and adherent cells on a culture dish are observed during DC preparation, differing in each patient. The harvesting of adherent cells from a culture dish by scraping inevitably causes a decrease in viability and recovery of the DC/monocyte ratio. It is reported that bone-marrow-derived DCs, which consist of non-adherent and adherent cells, may potentiate either tolerogenicity or pro-tumorigenic responses [32]. Heterogeneity of DCs leads to uncertainty of efficacy in cancer immunotherapy. Whereas the technology for the generation of non-adherent monocyte-derived DCs using non-adherent conditions have been previously established [33], the functional analysis of these DCs has not been sufficiently evaluated.

TAA peptides are loaded to activated DCs just before administration for clinical use [18,19,20,21,22,23,24]. Feuerstein et al. reported a method for producing DCs with preloaded tetanus toxoid or influenza matrix or melan-A antigen peptides before cryopreservation so they are ready to use after thawing [34]. Several groups performed a functional analysis of DC vaccines preloaded with antigen. The ability to stimulate T-cell proliferation by antigen-preloaded DCs has been well evaluated [35,36]; however, their induction of antigen-specific CTLs has not been fully determined.

Here, we developed DCs pre-pulsed with WT1 peptides in low-adhesion culture maturation (WT1-DCs). Quality tests (viability, phenotype, and function) of WT1-DCs were performed, and the findings were compared with those of conventional DCs (cDCs) prepared from adherent manufacturing systems. Furthermore, we evaluated the induction of WT1-specific CTLs in cancer patients who received WT1-DC administration.

## 2. Materials and Methods

### 2.1. Ethics and DC Preparation

#### 2.1.1. New Approach to Manufacture a DC Vaccine

DCs were generated in compliance with Good Gene, Cellular, and Tissue-based Products Manufacturing Practice. To generate a DC vaccine, processing was validated under the clinical study approved by the Ethical Committee of Kanazawa Medical University (approval number G131). A preclinical study was taken as an accompanying study of the DC vaccination therapy performed in patients with cancer. The DC vaccination study (approval number PC4160014, 10 June 2016) was approved by the Kanazawa Medical University Certificated Committee for Regenerative Medicine (Class III technologies) (approval number of the Committee NB4150006) according to the Act on the Safety of Regenerative Medicine introduced in Japan on 25 November 2014 [37], and all investigations were performed according to the Declaration of Helsinki.

Peripheral blood mononuclear cell (PBMC)-rich fraction was collected using leukapheresis with a Spectra Optia^®^ cell separator (Terumo BCT, Inc., Tokyo, Japan). PBMCs were subsequently isolated using a Ficoll-Plaque Premium (GE Healthcare, Piscataway, NJ, USA) density gradient. The collection and use of blood complied with relevant guidelines and institutional practices from Ethics Committees of Kanazawa Medical University. Written informed consent was obtained from all patients.

The antigenic profiles of the mature DCs (mDCs) were determined to be CD11c^+^, CD14^−^, human leukocyte antigen (HLA)-DR^+^, HLA-ABC^+^, CD80^+^, CD83^+^, CD86^+^, CD40^+^, and CCR7^+^ using flow cytometry [25]. The criteria for DC vaccine administration were as follows: purity (defined as >90% of CD86^+^HLA-DR^+^ cells), >70% viability, mDC phenotype, negative for bacterial and fungal infection after 14 days, presence of endotoxin ≤0.05 EU/mL, and negative for mycoplasma.

#### 2.1.2. DC Generation

Conventional DCs (cDCs) were generated using previously reported adhesion protocols [18,19,20,21,22,23,24]. Autologous PBMCs (2 × 10^7^) from patients were suspended in 6 mL of AIM-V medium (Thermo Fisher Scientific, Waltham, MA, USA) and placed into 100-mm adherent culture dishes (Primaria; BD Biosciences, San Jose, CA, USA). After removing non-adherent cells, 50 ng/mL of GM-CSF (Gentaur, Brussels, Belgium) and 50 ng/mL of IL-4 (R&D Systems, Inc., Minneapolis, MN, USA) were added the following day, and the cells were cultured for 5 days to generate immature DCs. Immature DCs were subsequently stimulated with a maturation cocktail containing 10 μg/mL of OK-432 (streptococcal preparation; Chugai Pharmaceutical Co., Ltd., Tokyo, Japan) and 50 ng/mL of prostaglandin E2, PGE2 (Daiichi Fine Chemical Co., Ltd., Toyama, Japan) for 24 h to generate mature DCs. Alternatively, WT1-DCs were prepared; PBMCs were again placed into adherent culture dishes (Primaria) in AIM-V medium. After removing non-adherent cells, 100 ng/mL of GM-CSF and 50 ng/mL of IL-4 (Miltenyi Biotec, Bergish Gladbach, Germany) were added the following day, and the cells were cultured for five days to generate immature DCs. These were subsequently stimulated with a maturation cocktail containing 10 μg/mL of OK-432, 10 ng/mL of PGE2 (Kyowa Pharma Chemical Co., Ltd., Toyama, Japan) and 20 μg/mL of the WT1 peptides reconstituted with DMSO (for WT1-235 killer peptide: CYTWNQMNL, residues 235–243; for WT1-34 helper peptide: WAPVLDFAPPGASAYGSL, residues 34–51; PEPTIDE INSTITUTE, INC., Osaka, Japan) for 24 h in low-attachment culture dishes (Prime Surface; Sumitomo Bakelite, Tokyo, Japan) to generate WT1-DCs.

### 2.2. Functional Analyses on the Obtained mDCs

#### 2.2.1. Phenotyping of DCs

Fluorescein isothiocyanate (FITC)- or phycoerythrin (PE)-conjugated monoclonal antibodies (mAbs) against the following DC markers were used: CD11c, CD80, CD86, PD-L1, PD-L2 or HLA-ABC (BD Pharmingen, San Diego, CA, USA); CD14, CD40, CD83, and HLA-DR (eBioscience, San Diego, CA, USA); and CD197 (R&D Systems). All analyses were performed on a FACSCalibur flow cytometer (BD Biosciences). After staining cells with each antibody, dead cells were removed by propidium iodide staining (Sigma-Aldrich, Steinheim, Germany), and live cells gated on forward scatter (FSC) and side scatter (SSC) without the lymphocyte population were examined for immunophenotyping.

#### 2.2.2. Pinocytotic and Phagocytic Assay

To evaluate pinocytotic or phagocytic activity, 100 µg/mL FITC-dextran (Molecular Probes, Eugene, OR, USA) for pinocytotic activity or 10 µg/mL DQ-ovalbumin (Molecular Probes) for phagocytic activity was added to the maturation cocktail. After the maturation process on DCs at 37 °C for 24 h, DCs were washed twice with FACS buffer and analyzed using flow cytometry.

#### 2.2.3. Measurement of Cytokine Production

Immature DCs were seeded at a density of 2 × 10^6^ cells/mL with maturation cocktail onto adherent or low-attachment 24-well plates. After maturation of DCs at 37 °C for 24 h, the collected supernatants were subsequently subjected to ELISA for IL-12p70, interferon (IFN)-γ, IL-10, and transforming growth factor (TGF)-β protein expression (R&D Systems) according to the manufacturers’ protocols.

#### 2.2.4. CTL Induction in Vitro

PBMCs from patients compatible with HLA-A*24:02 were used to generate mDCs. For post-pulsing with peptide, cryopreserved cDCs were thawed by heat block at 37 °C for 5 min and washed with saline. Then, cDCs were pulsed with 100 μg/mL of cytomegalovirus (CMV) peptide (QYDPVAALF, GenScript, Piscataway, NJ, USA) at 4 °C for 30 min. After washing cells twice with saline, cDCs were used as a stimulator. Alternatively, DCs pre-pulsed with CMV peptide in low-adhesion culture maturation (CMV-DCs) were thawed by heat block at 37 °C for 5 min, washed twice with saline, and subsequently used as the stimulator. CD8^+^T cells separated from HLA-A*24:02-autologous PBMCs using CD8 Microbeads (Miltenyi Biotec) were applied as responder cells. Stimulator (1 × 10^5^) and responder cells were co-cultured at a ratio of 1:10 in CTL medium supplemented with IL-2 (5 ng/mL; PeproTech, Rocky Hill, NJ, USA), IL-7 (5 ng/mL; R&D Systems), IL-15 (10 ng/mL; PeproTech), and 2-mercapto-ethanol (50 μg/mL; Bio-Rad Labs, Richmond, CA, USA). AIM-V media supplemented with 10% fetal bovine serum (Biosera, Dominican Republic) was added depending on cell expansion. After five days of cultivation, a half-medium change was performed by adding cDCs post-pulsed with CMV peptide or CMV-DCs in CTL medium. After three to five days of further incubation, the cells were harvested and 1 × 10^6^ cells were stained with FITC-conjugated anti-CD8 (Beckman Coulter, Inc., Brea, CA, USA) and APC-conjugated anti-CD3 (eBioscience) mAbs and T-select HLA-A*24:02 CMV pp65 Tetramer-QYDPVAALF (Medical and Biological Laboratories Co., Ltd., Nagoya, Japan) for analysis via flow cytometer. Dead cells were excluded by 7-AAD (BD Pharmingen) staining in flow cytometry analysis.

### 2.3. WT1-DC Administration

#### 2.3.1. Patients

Small numbers of patients with cancer were enrolled in the DC vaccination study to evaluate feasibility using a new WT1-DC vaccine after giving their informed consent. Vaccination was performed in combination with conventional chemotherapy for each patient. A DC vaccine was manufactured at the Regenerative Medicine Center, Kanazawa Medical University Hospital (FC4150228) and shipped to the outpatient clinic, Urata Clinic/SQOL Kanazawa for vaccination therapy. The DC vaccination study (approval number PC4180002, 23 April 2018) was approved by the Kanazawa Medical University Certificated Committee for Regenerative Medicine (Class III technologies) (approval number of the Committee NB4150006). The application requirements and conditions for the DC vaccination study were the same as a previous study [19]. The patients enrolled for DC vaccination had undergone rhG-CSF treatment 24–96 h prior to apheresis as described previously [25]. A total of seven patients with advanced cancers pathologically diagnosed as WT1 positive adenocarcinoma including stomach (three), colon/rectum (two), pancreatic (one), and salivary gland (one) cancers were enrolled; of those four patients compatible with HLA-A*24:02 were evaluated for immunological responses against WT1-CTLs.

#### 2.3.2. WT1-DC Administration

WT1-DCs were suspended in a total volume of 1 mL of saline containing 5% albumin (Japan Blood Products Organization, Tokyo, Japan), and 1–4 × 10^7^ WT1-DCs were injected at each time according to the number of DCs in each case for seven sessions (one course). The vaccine was intradermally and bilaterally administered near the axillary region and groin. DC vaccination was administered in seven sessions every two weeks following the protocol of DC vaccination [19].

#### 2.3.3. Immune Monitoring for WT1-CTLs

PBMCs were obtained before initiating the first vaccination and at the completion of the seventh vaccination. WT1 tetramer assay to detect WT1-CTLs was performed for patients who carried HLA-A*24:02. In total, 1 × 10^6^ PBMCs from patients who carried HLA-A*24:02 were stained with FITC-conjugated anti-CD8 (Beckman Coulter) and APC-conjugated anti-CD3 (eBioscience) mAbs, and T-Select HLA-A*24:02 modified WT1 Tetramer-CYTWNQMNL-PE or PE-conjugated HIV envelope/HLA-A*24:02 tetramer (Medical and Biological Laboratories Co., Ltd. MBL, Nagoya, Japan) served as a negative control. Dead cells were excluded via 7-AAD (BD Pharmingen) staining for flow cytometry. WT1 tetramer-positive CTLs were defined according to the following criteria: (1) comprising at least 0.02% of the CD3^+^CD8^+^ subset of 50,000–100,000 lymphocytes and (2) forming a clustered but not diffuse population [38].

Enzyme-linked immunospot (ELISpot) assays were performed using pre-coated human IFN-γ ELISpot PLUS kits (Mabtech, Nacka Strand, Sweden) to examine WT1-specific IFN-γ production by T cells referred to as CTLs. In total, 1 × 10^6^ PBMCs were seeded in 96-well plates in the presence of 10 μM WT1-235 killer peptides and WT1-34 helper peptides in AIM-V medium supplemented with 10% FBS. As a negative control, 10 μM HLA-A*24:02 HIV env (RYLRDQQLL, residues 584–592) (MBL, Nagoya, Japan), HLA-DRB1*01:01 HIV gag (DYVDRFYKTLRAE, residues 295–307; MBL, Nagoya, Japan), or DMSO was used. After 18–20 h of incubation, the emerged spots were calculated using an ELISpot reader (Autoimmun Diagnostika, Strassberg, Germany). Peptide-specific spots were enumerated by subtracting the spots of the control peptide from those of the WT1 peptides and expressed as the mean number of peptide-specific spots per 1 × 10^6^ PBMCs from duplicated wells. The presence of WT1-CTLs was defined according to the following criteria: (1) at least 15 WT1-specific spots per 1 × 10^6^ PBMCs and (2) at least 1.5-fold more WT1-specific spots than negative-control peptide spots [38].

#### 2.3.4. Shipping of WT1-DCs

Cryopreserved WT1-DCs were thawed by heat block at 37 °C for 5 min, washed twice with saline, and suspended in saline containing 5% albumin (Japan Blood Products Organization, Tokyo, Japan) before being enclosed in a tube. After packaging the tube with BARRIA POUCH (SUGIYAMA-GEN, Tokyo, Japan), the tube was shipped by BioBoxPLUS (SUGIYAMA-GEN) to the outpatient clinic, Urata Clinic/SQOL Kanazawa. A temperature range of 2 °C to 8 °C inside the BioBoxPLUS during shipping was monitored by a temperature data logger, TEMPRETRIEVER (MadgeTech, Warner, NH, USA).

### 2.4. Statistical Analysis

The Wilcoxon signed-rank test was used to compare differences among groups. All statistical analyses were performed using IBM SPSS Advanced Statistics software, version 23.0 (IBM Japan, Tokyo, Japan). Differences were considered statistically significant at a *p*-value < 0.05.

## 3. Results

### 3.1. WT1-DCs Show Remarkable Cluster and Increase in Viability and Recovery of DC/Monocyte Ratio Compared to Conventional DCs (cDCs)

In preparations of conventional DCs (cDCs) by using the adherent protocol, strong adherence to the culture dish decreases cell viability and recovery of the DC/monocyte ratio depending on the patient. For administration of cDC vaccines, cryopreserved cDC vaccines required post-pulsing with TAAs just prior to clinical use (Figure 1a, upper panel). Here, we developed a preparation of mature DCs pre-pulsed with WT1 peptides (WT1-DCs) in low-adherent conditions (Figure 1a, lower panel). After maturation stimulus with OK-432 (streptococcal preparation), PGE2, and WT1 peptides, floating cells were harvested by washing with medium, and cell morphology was observed by microscopy (Figure 1b, upper panel). Interestingly, remarkable floating non-adherent clusters were observed in WT1-DCs. Although cDCs resided in culture dishes after harvesting, almost no WT1-DCs adhered to the low-adherent culture dish (Figure 1b, lower panel). Compared with cDCs, WT1-DCs showed higher viability and recovery of the DC/monocyte ratio (Figure 1c; viability median: cDCs, 86%; WT1-DCs, 93%; yield median: cDCs, 27%; WT1-DCs, 30%), and analysis using a flow cytometer showed purity >70% in each DC vaccine (Purity median: cDCs, 81%; WT1-DCs, 81%).

### 3.2. Comparison of Phenotypes of WT1-DCs and cDCs

We found significant differences between WT1-DCs and cDCs adherent and low-adherent culture environments, as well as between either the presence or absence of antigen peptides in the DC maturation process (Figure 1). To determine the release criteria of WT1-DCs for vaccination, the phenotypes were analyzed by flow cytometry. The expression of CD11c, CD14, CD40, CD80, CD83, CD86, CD197 (CCR7), HLA-ABC, HLA-DR, PD-L1, and PD-L2 were analyzed (Figure 2). Compared with cDCs, WT1-DCs showed slightly higher expression of monocyte marker CD14 (median: cDCs, 0.94%; WT1-DCs, 2.2%), whereas expression of costimulatory molecule CD80, and immune checkpoint factor PD-L2 were lower compared with those of cDCs (CD80 median: cDCs, 86%; WT1-DCs, 76%; PD-L2 median: cDCs, 52%; WT1-DCs, 18%). Other cell surface antigens showed equivalent expression, and there was no significant difference between cDC and WT1-DCs in expression of CD86 or HLA-DR, which are the minimum criteria for DC vaccines (CD86 median: cDCs, 99%; WT1-DCs, 99%; HLA-DR median: cDCs, 99%; WT1-DCs, 99%). From these results, the release criteria of WT-DCs was defined as >90% of CD86+ and HLA-DR+ cells.

### 3.3. WT1-DCs Have Abilities of Lower Pinocytosis and IL-10 Production Compared with cDCs

To validate the potency of pre-pulsing of antigen and processing during maturation, pinocytosis and phagocytosis activities were examined during the maturation of DCs. Pinocytosis was observed by using FITC-dextran. Compared with cDCs, WT1-DCs showed slightly lower FITC Δ mean fluorescence intensity (ΔMFI) (median of ΔMFI: cDCs, 58; WT1-DCs, 54) (Figure 3, left panel). A lower pinocytosis activity was observed in WT1-DCs. In addition, analysis using DQ-ovalbumin, a self-quenched albumin that fluoresces upon proteolytic degradation, revealed that the fluorescence intensity generated from each DC was equivalent. These results indicated that the ability of cDCs and WT1-DCs to phagocytose was equivalent (Figure 3, right panel). Furthermore, the production of cytokines involved in the induction of CTLs was measured (Figure 4). Production of IL-12p70 and IFN-γ, which promote CTL induction, were equivalent. Despite a varying level based on each patient, WT1-DCs generated from three of seven showed higher IL-12p70 production than cDCs. The WT1-DCs also produced higher IFN-γ than cDCs. No change was observed in TGF-β secretion; however, a lower production of IL-10 was observed in WT1-DCs compared with cDCs (cDCs, 293 pg/mL; WT1-DCs, 39 pg/mL). Thus, compared with cDCs, WT1-DCs exhibited lower phagocytosis and IL-10 production.

### 3.4. Antigen-Presentation Ability of DCs Pre-Pulsed with CMV Peptide in Low-Adhesion Culture Maturation (CMV-DCs) are Similar to cDCs Post-Pulsed with CMV Peptide

To evaluate the antigen-presenting ability to activate CTLs, we prepared DCs pre-pulsed with CMV peptide in low-adhesion culture maturation (CMV-DCs). Compared with a culture of CD8^+^ T cells alone, co-culture of CD8^+^ T cells with cDCs post-pulsed with CMV peptide or CMV-DCs resulted in a marked increase in CMV-specific CTLs (Figure 5, upper panel; CD8^+^ T cells, 0.1%; CD8^+^ T cells + cDCs post-pulsed with CMV, 9.9%; CD8^+^ T cells + CMV-DCs, 9.1%). There was no significant difference in the ratio of CMV-specific CTLs induced by cDCs post-pulsed with CMV and CMV-DCs (median: cDCs post-pulsed with CMV, 5.9%; CMV-DCs, 6.3%) (Figure 5, lower panel).

### 3.5. Administration of WT1-DCs Induces WT1-Specific CTLs in Patients with Cancer

As an interim analysis, four patients having HLA-A*24:02 received the WT1-DCs vaccine were evaluated, which had been shipped to the neighboring clinic within 1 h after release. Immunohistochemistry was also performed for WT1 antigens on paraffin embedded tissues before enrolling the study (data not shown). The condition of all vaccines met the administration criteria. Immune monitoring using tetramer analysis and ELISpot assays were performed after one course of the DC vaccination. Of the four patients completing one course of WT1-DCs vaccination three had gastric cancer and one had salivary gland cancer. Pre-DC vaccination status, post-DC vaccination status, and immunological responses are shown in Table 1; Table 2. WT1-CTLs from three male patients with gastric cancer were detected using WT1-tetramer analysis (Figure 6). The detection of IFN-γ-producing cells showed elevation after WT1-DCs vaccination in three patients using ELISpot assays (Figure 6). Conversely, in patient No.4 with salivary gland cancer, the immunological responses failed as negative following to the criteria of immune monitoring for WT1-CTLs [38].

In patient No. 3 (upper panel in Figure 6), the induction of WT1-CTLs was observed via tetramer analysis (Before Vac., 0.01%; After Vac., 0.10%). Despite the increased level of WT1 peptides according to ELISpot assays, the non-specific elevation of IFN-γ-producing cells was found in the control stimulation after one course of WT1-DC vaccination. Therefore, the specificity for detecting WT1-CTLs could not be confirmed according to the previously reported criteria [39]. In patient No. 5 (middle panel in Figure 6), an increased number of WT1-CTLs was detected after WT1-DC vaccination (Before Vac., 0.00%; After Vac., 0.05%). Conversely, a slight increase in the number of spots containing WT1 peptides was observed after one course of vaccination, the number of spots containing control peptides also increased similarly. Patient No. 6 (lower panel in Figure 6) received the WT1-DCs vaccine under this protocol but had also undergone cDCs vaccination study using the previous protocol with post-pulsed WT1 peptides [approval number PC4160014, June 10, 2016]. In this case, the number of WT1-CTLs increased after the second course of DC vaccination (before second Vac., 0.16%; after second Vac., 0.19%) on the positive baseline of both tetramer analysis and ELISpot assays. The number of IFN-γ spots further increased after the second course of DC vaccination pulsed with WT1-235 killer and WT1-34 helper peptides compared with that after the first session without additional chemotherapy.

## 4. Discussion

In this study, we performed phenotypic and functional analyses on WT1-DCs pre-pulsed with WT1 peptides in low-adhesion culture maturation, and we evaluated active WT1-CTLs after WT1-DC administration in patients with cancer. Compared with cDCs, WT1-DCs formed floating clusters and increased in viability and recovery of the DC/monocyte ratio. By contrast, co-stimulatory molecule CD80 and the immune checkpoint factor PD-L2 on WT1-DCs expressed lower levels than those on cDCs. In addition, the production of immune suppressive cytokine IL-10 from WT1-DCs was extremely low. Nevertheless, different DC maturation protocols (cDC vs. WT1-DC) did not affect antigen-presentation ability. Furthermore, immune monitoring of WT1-CTLs as practical application of the total process including shipping of the WT1-DC vaccine after WT1-DC-administration demonstrated that WT1-DCs induced WT1-CTLs in patients with cancer.

In the conventional adherent protocol for monocyte-derived DC generation, adherence of cells to the culture plate was dependent on the patient from whom the cells originated (Figure 1b). Recovery of adherent cells by scraping causes a decrease in cell viability and recovery of the DC/monocyte ratio yield. WT1-DCs generated from a low-adhesion dish were easy to recover by washing with medium and showed a high viability and recovery. Low expression of DC-specific intercellular adhesion molecule-3-grabbing non-integrin (DC-SIGN, also called CD209) and PD-L2 has been reported for these populations [33]. DC-SIGN is involved in antigen uptake [39]; thus, it is speculated that the reduction of pinocytosis might have been caused by low DC-SIGN expression on WT1-DCs. However, WT1-DCs have an equivalent capacity for antigen presentation as cDCs (Figure 5). The difference of pinocytosis between cDCs and WT1-DCs did not affect CTL induction in vitro.

Our phenotypic analysis of WT1-DCs also showed that CD80 and PD-L2 expression were significantly reduced compared to cDCs. The streptococcal preparation OK-432 engages TLR2 or TLR4 [29] and promotes maturation of human monocyte-derived DCs correlated with increased expression of CD80, CD83, and CD86 [29,30,31]. The downregulation of CD80 and PD-L2 together with the remarkable cluster formation that occurs with WT-DCs might therefore reduce the signaling of OK-432 via TLRs and affect the maturation. We expected to produce non-adherent DCs equipped with homogeneous in phenotype and function by using the optimized manufacturing protocol of WT-DCs. WT1-DCs exhibited heterogeneous phenotype and function, the control of cluster formation may be an important issue for the progress of homogeneous WT1-DCs. Size-dependent hepatic differentiation of human induced pluripotent stem (iPS) cells has been reported [40]. The control of cell mass size is important for the efficiency and reproducibility of differentiation of functional cells from iPS cells. The regulation of cluster size of DCs could contribute to generating homogenous DC vaccines equipped with the ability to induce high acquired immunity.

Compared with cDCs, WT1-DCs showed an equivalent production of IL-12p70, IFN-γ, and TGF-β but a low production of the immunosuppressive cytokine IL-10. IL-10 production from antigen-presenting cells is specific for TLR2 agonists [41,42,43,44]. We speculate that insufficient OK-432 signaling through TLR2 might led to a decrease in the IL-10 production of WT1-DCs. *Mycobacterium avium* induces PD-L2 expression on mouse bone marrow-derived dendritic cells in an IL-10-dependent manner via the TLR2-p38-MAPK signaling pathway [45]. Therefore, the low PD-L2 expression observed in WT1-DCs may be due to a reduction in IL-10 production, but further study is needed to test this.

Knockdown of PD-L1 and PD-L2 in monocyte-derived DCs enhances CTL induction [46], and IL-10 has known immunosuppressive effects [47]. Therefore, reduced PD-L2 and IL-10 in WT1-DCs were expected to enhance their induction of CTLs. However, in vitro CTL induction revealed that antigen-presentation abilities were equivalent in cDCs and WT1-DCs (Figure 5). Nevertheless, some suppression of PD-L1 and PD-L2 expression is needed to enhance CTL induction. DC vaccines with siRNA silencing of PD-L1 and PD-L2 augment the expansion and function of CD8^+^ T cells specific for minor histocompatibility antigens [48]. Indeed, clinical trials for hematological malignancies using DCs with siRNAs against PD-L1 and PD-L2 (NCT02528682) are expected to lead to the development of promising DC vaccines.

The acquisition of WT1-CTLs as a proof-of-concept drug delivery in vivo was observed in patients with cancer who received WT1-DCs vaccination. The effectiveness of WT1-DCs to induce acquired immunity was confirmed. Specifically, IFN-γ production was negative in two of three cases treated with WT1-DCs (Table 1, Table 2 and Figure 6). It is important to deliberate the possible reason why WT1-CTLs showed IFN-γ negativity to understand its antitumor activity. WT1-DCs vaccination was conducted without any adjuvants in this study, which may have led to a failure in the induction function in vivo. Administration of WT1-DCs with OK-432 might be essential for achieving sufficient induction of functional WT1-CTLs in patients with cancer. In fact, the induction of IFN-γ producing WT1-CTLs was observed after the vaccination with WT1-post-pulsed DCs in combination with OK-432 in vivo [19,49]. It has been reported that OK-432 induces IL-12 production from human PBMCs and promotes a Th1 dominant state that is suitable for inducing antitumor immunity [50,51]. Moreover, OK-432 significantly enhanced in vitro proliferation of CD4^+^ effector T cells by regulatory T (Treg)-cell suppression, and this blocking effect depended on IL-12 derived from antigen-presenting cells [52]. The induction of IFN-γ producing WT1-CTLs without an increase in Treg cells was observed after the administration of WT1-post-pulsed DCs with OK-432 [53]. Several preclinical and clinical studies suggest that Treg cells prevent the development of effective antitumor immunity in tumor-bearing patients and promote tumor progression [54]. The activity of OK-432 to Treg-cell suppression could be beneficial for the induction of functional WT1-CTLs in vivo. Further clinical studies using WT1-pre-pulsed DCs with OK-432 for patients with cancer would be needed to monitor the induction of IFN-γ producing WT1-CTLs as wells as to improve the immune environment in vivo.

## 5. Conclusions

In conclusion, we established a protocol for the preparation of WT1-DCs pre-pulsed with WT1 peptides in optimized culture maturation. WT1-DCs exhibit high viability, recovery, and equivalence in in vitro CTL induction compared with cDCs. After the administration of WT1-DCs, immune monitoring demonstrated that WT1-DCs induce acquired immunity in patients with cancer. DCs function as adjuvants in vivo and are expected to be applied to cancer treatments that promote long-lasting effects with few side effects. WT1-DC vaccination for patients with cancer demonstrated the safety and immunogenicity in vivo. Prospective clinical trials are required to evaluate the efficacy of acquired immunity in response to WT1-DC vaccination in large number of cancer patients.

## 6. Patents

S.S. and T.K. are inventors of the patent for the manufacturing of a DC vaccine using G-CSF (PCT/JP/2014/053676). H.S. is the inventor of the WT1 patent (PCT/JP2010/057149 and PCT/JP2006/323827).

## Figures and Tables

**Figure 1 pharmaceutics-12-00305-f001:**
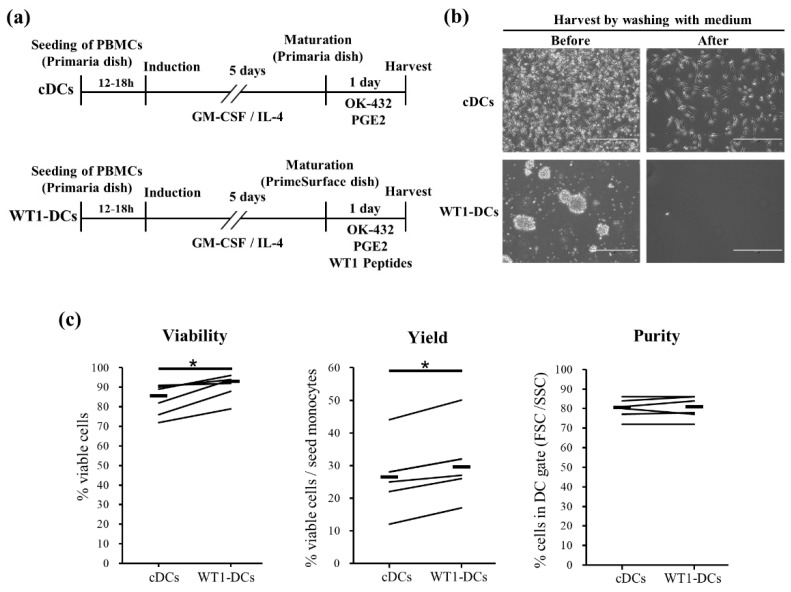
DCs pre-pulsed with Wilms’ tumor (WT1) peptides in low-adhesion culture maturation (WT1-DCs) form remarkable floating clusters and show higher viability and recovery of the DC/monocyte ratio. (**a**) In the preparation of conventional DCs (cDCs) by using the conventional adherent protocol, immature DCs were suspended with mature medium containing OK-432 and PGE2 and seeded on an adherent culture dish. After 24 h cultivation, floating and loosely attached cells were collected by washing with medium and strongly attached cells were collected by scraping. Alternatively, for the preparation of WT1-DCs, immature DCs were suspended with mature medium containing OK-432, PGE2, and WT1 peptides, seeded on a low-adherent culture dish, and harvested by washing with medium after 24 h. (**b**) Observation of cells using phase-contrast microscopy before and after harvesting by washing with medium. White bar indicates 400 μm. (**c**) Live and dead cells were measured by trypan blue staining for comparison of viability and recovery of the DC/monocyte ratio. Purity of DCs was measured by flow cytometer. PI-negative and gated cell population from FSC and SSC, excluding lymphocytes, were defined as DCs (*n* = 6). * *p* < 0.05.

**Figure 2 pharmaceutics-12-00305-f002:**
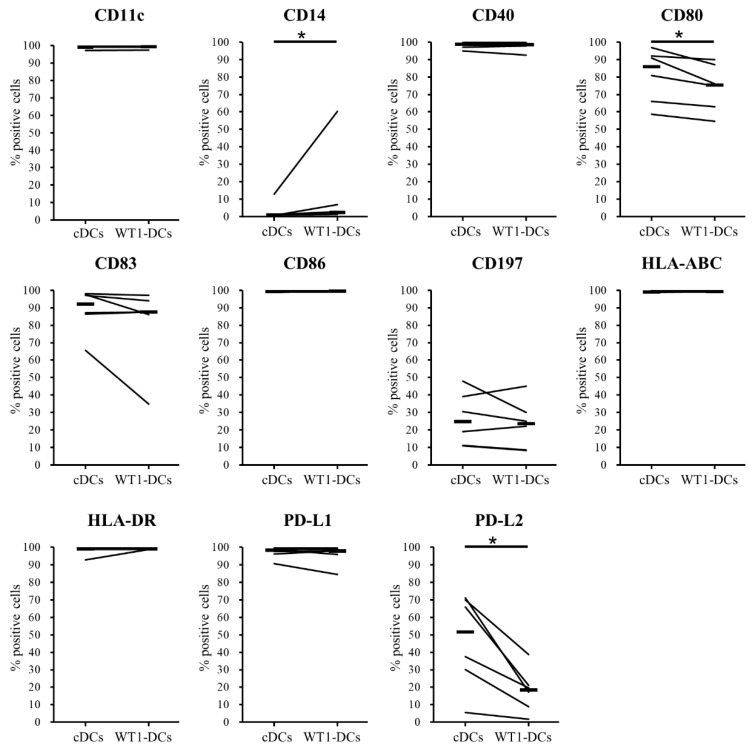
Comparison of dendritic cell (DC) phenotypes. After harvesting cDCs and WT1-DCs prepared from the same donors, DCs were stained with antibodies for DC markers and analyzed using a flow cytometer (*n* = 6). The population of positive cells was determined in propidium iodide (PI)-negative and DC-gated populations excluding lymphocytes from forward and side scatter. * *p* < 0.05.

**Figure 3 pharmaceutics-12-00305-f003:**
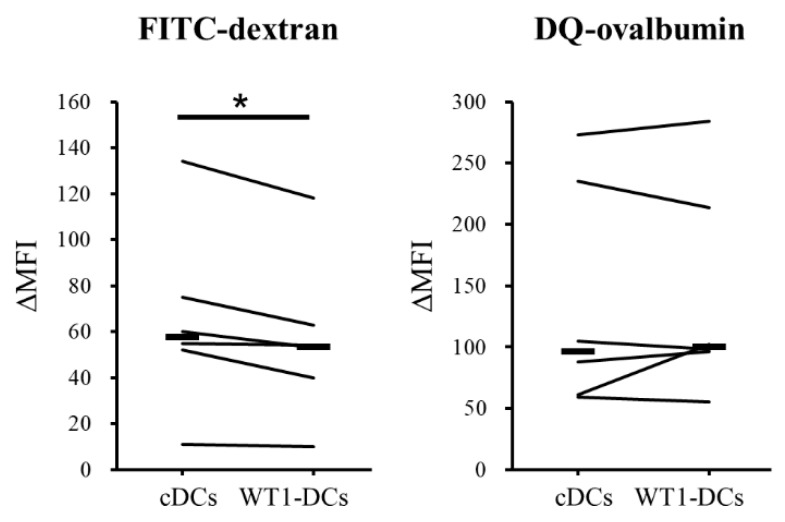
Comparison of pinocytotic and phagocytic activities. DCs were incubated with FITC-dextran for antigen pinocytotic or DQ-ovalbumin for antigen phagocytic activities with maturation cocktail. These cells were washed after a 24 h incubation, and the fluorescence intensity was examined by flow cytometer (*n* = 6). Δ mean fluorescence intensity (MFI) indicates a value obtained by subtracting the control incubated with DMSO. * *p* < 0.05.

**Figure 4 pharmaceutics-12-00305-f004:**
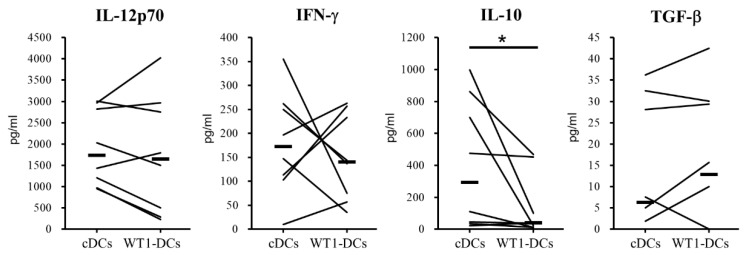
Comparisons of cytokine production from cDCs or WT1-DCs. The culture supernatant after maturation of DCs were subjected to measuring of cytokine production. The amount of IL-12p70, IFN-γ, IL-10 and TGF-β were determined by ELISA (*n* = 8). * *p* < 0.05.

**Figure 5 pharmaceutics-12-00305-f005:**
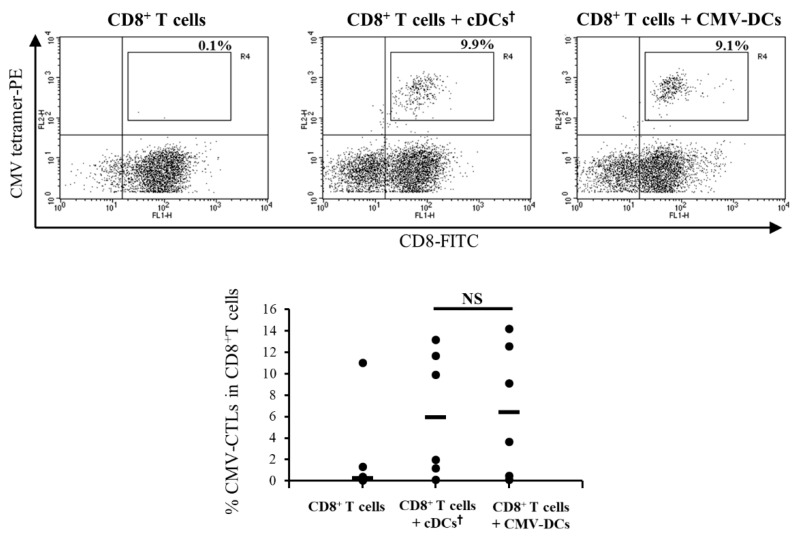
cDCs post-pulsed with cytomegalovirus (CMV) peptide and DCs pre-pulsed with CMV peptide in low-adhesion culture maturation (CMV-DCs) show equivalent antigen-presenting abilities. Representative data of CMV-specific cytotoxic T lymphocytes (CTLs) induced by cDCs post-pulsed with CMV peptide or CMV-DCs (upper panel). The cultivation of CD8^+^ T cells only was the negative control. The percentage in each panel indicates the ratio of CMV-tetramer^+^ CTLs in CD8^+^ T cells. The lower panel shows a summary of CMV-specific CTLs induction in CD8^+^ T cells (*n* = 6). †, post-pulsed with CMV peptide. NS, not significant.

**Figure 6 pharmaceutics-12-00305-f006:**
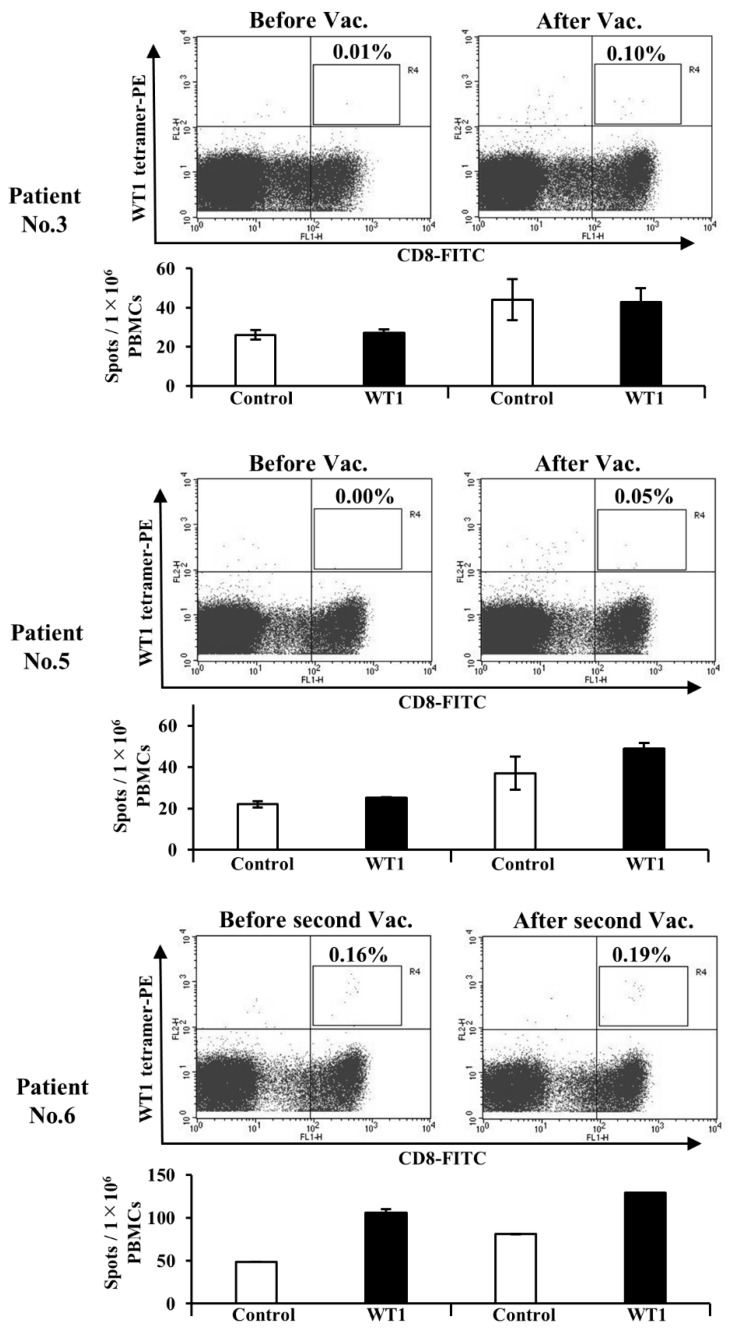
Immune monitoring of WT1-specific CTLs (WT1-CTLs) in patients after WT1-DC administration. Patients No. 3 and No. 5 received one course of WT1-DC vaccination. Thereafter, the induction of WT1-CTLs was detected via WT1 tetramer analysis, and IFN-γ release from WT1-CTLs was assessed using Enzyme-linked immunospot (ELISpot) assays. Patient No. 6 received a DC vaccine post-pulsed with WT1 peptides using the previous protocol. After the second course of administration of the WT1-DC vaccine, the maintenance of WT1-CTL function was evaluated. The percentages in the dot plot panels show the ratio of WT1-CTLs in CD8^+^ T cells. The opened or closed bars indicate the numbers of spots from PBMCs stimulated with control or WT1 peptides, respectively. The mean number of spots from duplicate wells is shown.

**Table 1 pharmaceutics-12-00305-t001:** Clinical characteristics of patients treated with the WT1-pulsed DC vaccine.

Patient No.	Age (Years)	Sex	Disease	Pre-DC Vaccination Status		
				Therapy	Stage (Operation)	Chemotherapy
3	60	M	Gastric cancer	post operation; post chemotherapy	II	S1
4	57	M	Salivary gland cancer	post operation; post radiation; post chemotherapy	IV	CDDP, Trastuzumab
5	58	M	Gastric cancer	post operation; during chemotherapy	IV	XELOX
6	68	M	Gastric cancer	post chemotherapy; post operation; DC vaccination	IA	non

Abbreviations: DC(s), dendrititic cell(s); S1, Tegafur/gimeracil/oteracil; CDDP, cisplatin; XELOX, Capecitabine + oxaliplatin; WT1, Wilms’ tumor 1.

**Table 2 pharmaceutics-12-00305-t002:** Immune monitoring of WT1-CTLs in patients treated with the WT1-pulsed DC vaccine.

Patient	HLA Typing						DC Vaccination			Immunological Responses	
No.	A		DR		DQ		Status (Stage)	Total No. of DCs (×10^7^)	Combination chemotherapy	ELISPOT *	Tetramer
3	2402	-	0405	0901	0201	-	CR	12	non	negative	positive
4	0301	2402	0403	1301	0201	-	IV	8	non	negative	negative
5	2402	2601	0901	1101	0201	-	IV	9	XELOX	negative	positive
6	1101	2402	0405	-	0501	-	IV	7	non	positive	positive

Abbreviations: DC(s), dendritic cell(s); S1, Tegafur/gimeracil/oteracil; XELOX, Capecitabine + oxaliplatin; ELISPOT, enzyme-linked immunospot assay; WT1, Wilms’ tumor 1; * the IFN-γ production from WT1-CTLs was defined according to the criteria [38].

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
