# Peer review of "Dendritic Cells Pre-Pulsed with Wilms’ Tumor 1 in Optimized Culture for Cancer Vaccination"

_pharmaceutics, 2020, doi:10.3390/pharmaceutics12040305_

Round 1

Reviewer 1 Report

The authors claim to have come up with a method to generate an "innovative application of low-adhesion culture maturation of dendritic cells for cancer vaccination" however the design of the study suffers major flaws.

Indeed, the authors compare head to head the DC generated with two methods which have 3 variables (one method use 50ng/mL GM-CSF the other uses 100ng/mL then one method add WT1-derived peptide the other add no peptide and finally one method use adherent petri dish for the maturation step while the other employs a low-adherent petri dish), yet the author attribute all the differences observed to the low versus high adherence petri dishes used during the maturation process. The method described does not state how many PBMC were used nor the volume of media used. Nothing is written on where the peptides were purchased and nor how they were reconstituted. Where was the tetramer purchased, how were all the staining performed?

The abilities of the DC to pinocytose should have been assessed before and after maturation not while the cells were cultured in the presence of maturating agents. Can the authors explain why they are using OK-432 to mature their DC?

Further more the authors have enrolled 7 patients with advanced cancers to receive the DC-WT1 vaccination without first checking whether the tumour of these patients express WT1, the vaccine uses an HLA-A24 WT1 derived peptide and yet the HLA-A haplotype of the patients was not checked before enrolling the patients. In addition, the patients received different number of DC. Only 3 patients at the end received the vaccine which is a very low number of patients. No ELISPOT data are shown.

The entire paper is not clear and need major revisions and more patients.

Author Response

We thank you greatly for your e-mail and review our manuscript (Manuscript ID

pharmaceutics-720810) that we submitted on January 31, 2020. We appreciate the constructive comments provided by the reviewers that have allowed us to substantially improve our manuscript.

We are submitting a revised version of the manuscript. All changes have been made in response to the reviewers’ individual comments.

Reviewer #1:

Comments and Suggestions for Authors

The authors claim to have come up with a method to generate an "innovative application of low-adhesion culture maturation of dendritic cells for cancer vaccination" however the design of the study suffers major flaws.

Indeed, the authors compare head to head the DC generated with two methods which have 3 variables (one method use 50ng/mL GM-CSF the other uses 100ng/mL then one method add WT1-derived peptide the other add no peptide and finally one method use adherent petri dish for the maturation step while the other employs a low-adherent petri dish), yet the author attribute all the differences observed to the low versus high adherence petri dishes used during the maturation process.

Answer) We appreciate your kind comments and excellent suggestions. We aimed to compare DCs pre-pulsed with Wilms tumor 1 (WT1) peptides in low adhesion culture maturation (WT1-DCs) as a cancer vaccine with conventional DCs (cDCs) under process validation. We have added the words “for process validation” in line 30. We did not verify the effect of differences of the GM-CSF concentration, peptide pulsing, or high or low adherent maturation conditions. To avoid confusion, the title has been revised to “Dendritic cells pre-pulsed with Wilms' tumor 1 in optimized culture for cancer vaccination,” and phrasing related to differences in DCs associated with low-adhesion culture or the presence or absence of the peptide and GM-CSF concentration were deleted from the original manuscript.

The method described does not state how many PBMC were used nor the volume of media used.

Answer) We have added the relevant information on lines 123–125.

Nothing is written on where the peptides were purchased and nor how they were reconstituted.

Answer) We have added the phrase “reconstituted with DMSO” in line 136 and the supplier information “PEPTIDE INSTITUTE, INC., Osaka, Japan” in line 138.

Where was the tetramer purchased, how were all the staining performed?

Answer) We have added the relevant information in lines 207–214.

The abilities of the DC to pinocytose should have been assessed before and after maturation not while the cells were cultured in the presence of maturating agents.

Answer) We clarified this information in lines 290–291, and added a sentence in the Discussion in lines 391–394.

Can the authors explain why they are using OK-432 to mature their DC?

Answer) We added the sentence in lines 69–79.

Further more the authors have enrolled 7 patients with advanced cancers to receive the DC-WT1 vaccination without first checking whether the tumour of these patients express WT1, the vaccine uses an HLA-A24 WT1 derived peptide and yet the HLA-A haplotype of the patients was not checked before enrolling the patients.

Answer) We have clarified this issue in lines 184–185, 194–195, and 196–197.

 In addition, the patients received different number of DC. Only 3 patients at the end received the vaccine which is a very low number of patients. No ELISPOT data are shown.

Answer) We explained why patients received different numbers of DCs in lines 199–201. The method of the ELISpot assay has been described on lines 217–227. The results of WT1 tetramer and ELISpot assays have been added as Figure 6 and described in lines 334–359.

The entire paper is not clear and need major revisions and more patients.

Answer) In this study, we developed a DC vaccine pre-pulsed with WT1 peptides in optimized culture for cancer vaccination. We performed functional analyses on the WT1-DC vaccine products from patients with cancer to achieve process validation of vaccine lots. Thereafter, we evaluated the feasibility of treatment with WT1-DCs in patients with cancer who carry the HLA-A*24:02 mutation using validated immune monitoring analyses, obtaining detectable WT1-specific CTLs. Taken together, a standardized protocol for pre-pulsed WT1-DCs might be useful for cell-based drug delivery systems, as described in the revised Abstract. We have also added the following sentences to the Conclusion section: “WT1-DC vaccination for patients with cancer demonstrated the safety and immunogenicity in vivo. Prospective clinical trials are required to evaluate the efficacy of acquired immunity in response to WT1-DC vaccination in large number of cancer patients.”

Reviewer 2 Report

This research group generated ex vivo monocyte-derived DCs and developed WT1 peptide encapsulated mature DCs under a low-adherent condition. The low-adherent condition during DC maturation showed enhanced DC viability, better recovery of the DC/monocyte ratio, lower IL-10 display, similar antigen-presenting ability, and CTL induction compared to conventional adherent DC maturation system. The concept of the research was innovative and could provide promising advanced DC therapy for cancer patients, but there were some concerns in experimental designs and details.

The authors compared two DCs; cDCs and WT1-DCs. Why the cDCs have not been treated with WT-1 peptides during the DC maturation? To compare the conventional method and new method, two DCs should have been treated same. The DCs matured by new method (floating maturation) were treated with WT-1 peptides, it was hard to know the differences between the conventional and new methods were from peptides or maturation system. The authors need to add the right control experiments or explain the experimental design carefully. Generally, antigen uptake of DC decreases after maturation. Therefore, it was hard to get meaningful information from figure 3. After FITC-dextran treatment, WT1-DCs showed reduced Ag uptake compared to cDCs. This result can be explained two ways; 1) The new method (floating maturation) enhanced DC maturation. Or, 2) WT-1 peptide treatment during maturation induced more DC maturation. Again, without a correct control experiment, it is hard to conclude the data. In figure 4, the authors compared cytokine production between cDCs and WT1-DCs. In case of IL-12p70, IFN-g, and TNF-beta, there were many variations in individuals. Some showed higher cytokine levels in WT1-DCs, but some showed higher cytokine levels in cDCs. I am wondering if each cytokine production of each individuals had same patterns or not. For example, if one patient’s DCs produced higher IL-12p70 in WT1-DCs than those in cDCs, this DCs also produced higher IFN-g and TNF-beta in WT1-DCs than cDCs? And why the individuals presented various DC functions even the immature DCs were generated from monocytes? Seven patients were received the WT1-DC vaccine, but only 3 patients’ data was shown. Is there any specific reason to exclude 4 patients’ data?

Author Response

We thank you greatly for your e-mail and review our manuscript (Manuscript ID

pharmaceutics-720810) that we submitted on January 31, 2020. We appreciate the constructive comments provided by the reviewers that have allowed us to substantially improve our manuscript.

We are submitting a revised version of the manuscript. All changes have been made in response to the reviewers’ individual comments.

Reviewer #2:                                                            

This research group generated ex vivo monocyte-derived DCs and developed WT1 peptide encapsulated mature DCs under a low-adherent condition. The low-adherent condition during DC maturation showed enhanced DC viability, better recovery of the DC/monocyte ratio, lower IL-10 display, similar antigen-presenting ability, and CTL induction compared to conventional adherent DC maturation system. The concept of the research was innovative and could provide promising advanced DC therapy for cancer patients, but there were some concerns in experimental designs and details.

The authors compared two DCs; cDCs and WT1-DCs. Why the cDCs have not been treated with WT-1 peptides during the DC maturation? To compare the conventional method and new method, two DCs should have been treated same. The DCs matured by new method (floating maturation) were treated with WT-1 peptides, it was hard to know the differences between the conventional and new methods were from peptides or maturation system. The authors need to add the right control experiments or explain the experimental design carefully.

Answer) We appreciate your kind comments and excellent suggestions. We aimed to compare DCs pre-pulsed with Wilm’s tumor (WT1) peptides in low adhesion culture maturation (WT1-DCs) as a cancer vaccine with conventional DCs (cDCs) under process validation. We have added the words “for process validation” in line 30. We did not verify the effect of differences of peptides and the maturation system. To avoid confusion, the title has been revised to “Dendritic cells pre-pulsed with Wilms' tumor 1 in optimized culture for cancer vaccination,” and phrasing related to differences in DCs associated with the low-adhesion culture, the presence or absence of the peptide, and the GM-CSF concentration were deleted from the original manuscript.

Generally, antigen uptake of DC decreases after maturation. Therefore, it was hard to get meaningful information from figure 3. After FITC-dextran treatment, WT1-DCs showed reduced Ag uptake compared to cDCs. This result can be explained two ways; 1) The new method (floating maturation) enhanced DC maturation. Or, 2) WT-1 peptide treatment during maturation induced more DC maturation. Again, without a correct control experiment, it is hard to conclude the data.

Answer) We have clarified this information in lines 290–291 of the revised manuscript and also added a sentence in the Discussion section (lines 391–394).

In figure 4, the authors compared cytokine production between cDCs and WT1-DCs. In case of IL-12p70, IFN-g, and TNF-beta, there were many variations in individuals. Some showed higher cytokine levels in WT1-DCs, but some showed higher cytokine levels in cDCs. I am wondering if each cytokine production of each individuals had same patterns or not. For example, if one patient’s DCs produced higher IL-12p70 in WT1-DCs than those in cDCs, this DCs also produced higher IFN-g and TNF-beta in WT1-DCs than cDCs?

Answer) We have added information regarding this in lines 299–301 of the revised manuscript.

And why the individuals presented various DC functions even the immature DCs were generated from monocytes?

Answer) We have addressed this issue in lines 123–125 of the revised manuscript. Because DCs were generated from autologous patient PBMCs, the individual viability, yield, cell surface phenotype, and function were clarified, as presented in Figures 1c, 2, 3, 4, and 5.

Seven patients were received the WT1-DC vaccine, but only 3 patients’ data was shown. Is there any specific reason to exclude 4 patients’ data?

Answer) We have clarified this issue in lines 180–181, 190–191, and 192–193. The results of WT1 tetramer and ELISpot assays have been added as Figure 6 and described in lines 329–355.

Reviewer 3 Report

The authors demonstrate the isolation, maturation, antigen-priming, and utilization of human monocyte-derived dendritic cells using a low-adhesion culturing protocol. The work included in vitro work using human cells and a small in vivo study with human subjects.

The writing of the manuscript is of very good quality and overall it was easy to understand. One of the only noticeable issues was the use of the phrase "internally capsuled," which I do not recognize as a term of art used in the field of immunology. "Internally capsuled" was used 8 times in the manuscript, each time describing the low-adhesion dendritic cells which has been preloaded with antigen prior to cryogenic preservation. Indeed, the authors cite REF 25, which refers to such dendritic cells as "antigen-preloaded mature dendritic cells," a more intelligible description. If the authors would like to introduce "internally capsuled" to the field, it would be best if they define the term in the introduction.

The authors end their abstract with a mention of wearable/implantable devices, but I do not recall a single instance in the manuscript where this idea is revisited.

An additional issue of the paper is inadequate description of the low-adhesion surface and the choice thereof. While the introduction briefly mentions REF 24 and the potential benefits of low-adhesion maturation of dendritic cells, the authors do not further describe their rationale for the particular surface the chose. This would help readers evaluate the vetting of surfaces and aid readers in the selection of their own surfaces.

The meaning of lines 47-50 in the introduction is not clear. To begin with, incomplete Freund's adjuvant and Montanide ISA 51 are one and the same, and therefore the sentences make little sense (https://www.cancer.gov/publications/dictionaries/cancer-terms/def/montanide-isa-51). It is not clear how the findings of REF 11 support the following sentence: "Therefore, if a peptide vaccine’s immunogenic potency tends to be weak, appropriate adjuvants and/or delivery systems may be used to potentiate this." Nor is it clear why a discussion of adjuvants is actually relevant to the topic of this paper. Perhaps the authors could provide clarification as to their thought process resulting in the inclusion of this topic.

Authors should make more clear that IFN-y ELISPOT negative results are indicative of a lack of activation of T cells, suggesting that the dendritic cells were not effective at activating T cells in vivo. The positive qualities of low-adhesion maturation (slightly increased viability, slightly increased yield, decreased PD-L1, decreased IL-10) stand in sharp contrast to the lack of in vivo efficacy -- something the authors must address more strongly in the discussion.

It is also confusing that 7 patients were enrolled but results are only displayed for 3 patients. Are the authors going to display results for the other 4 patients, or at least of all 4 HLA-A*24:02. 3/4 of these patients were tetramer positive, but only 1/4 were IFN-y positive. 

In terms of citations, many of the dendritic cell cancer therapies cited are self-referential, and some thought should be put into citing other high profile advances in the field, such as https://stm.sciencemag.org/content/10/436/eaao5931?et_cid=1967027&et_rid=314188761.

Author Response

We thank you greatly for your e-mail and review our manuscript (Manuscript ID

pharmaceutics-720810) that we submitted on January 31, 2020. We appreciate the constructive comments provided by the reviewers that have allowed us to substantially improve our manuscript.

We are submitting a revised version of the manuscript. All changes have been made in response to the reviewers’ individual comments.

Reviewer #3:                      

The authors demonstrate the isolation, maturation, antigen-priming, and utilization of human monocyte-derived dendritic cells using a low-adhesion culturing protocol. The work included in vitro work using human cells and a small in vivo study with human subjects.

The writing of the manuscript is of very good quality and overall it was easy to understand. One of the only noticeable issues was the use of the phrase "internally capsuled," which I do not recognize as a term of art used in the field of immunology. "Internally capsuled" was used 8 times in the manuscript, each time describing the low-adhesion dendritic cells which has been preloaded with antigen prior to cryogenic preservation. Indeed, the authors cite REF 25, which refers to such dendritic cells as "antigen-preloaded mature dendritic cells," a more intelligible description. If the authors would like to introduce "internally capsuled" to the field, it would be best if they define the term in the introduction.

Answer) We appreciate your insightful comment. “Internally capsuled” has been revised to “pre-pulsed” in line 28.

The authors end their abstract with a mention of wearable/implantable devices, but I do not recall a single instance in the manuscript where this idea is revisited.

Answer) We have revised “implantable/wearable medical device” to “cell-based drug delivery system” in line 37.

An additional issue of the paper is inadequate description of the low-adhesion surface and the choice thereof. While the introduction briefly mentions REF 24 and the potential benefits of low-adhesion maturation of dendritic cells, the authors do not further describe their rationale for the particular surface the chose. This would help readers evaluate the vetting of surfaces and aid readers in the selection of their own surfaces.

Answer) We have addressed this issue in lines 400–403.

The meaning of lines 47-50 in the introduction is not clear. To begin with, incomplete Freund's adjuvant and Montanide ISA 51 are one and the same, and therefore the sentences make little sense (https://www.cancer.gov/publications/dictionaries/cancer-terms/def/montanide-isa-51).

Answer) We have added the termMontanide ISA51” in line 49.

It is not clear how the findings of REF 11 support the following sentence: "Therefore, if a peptide vaccine’s immunogenic potency tends to be weak, appropriate adjuvants and/or delivery systems may be used to potentiate this." Nor is it clear why a discussion of adjuvants is actually relevant to the topic of this paper. Perhaps the authors could provide clarification as to their thought process resulting in the inclusion of this topic.

Answer) We have clarified this issue in lines 51–53.

Authors should make more clear that IFN-y ELISPOT negative results are indicative of a lack of activation of T cells, suggesting that the dendritic cells were not effective at activating T cells in vivo. The positive qualities of low-adhesion maturation (slightly increased viability, slightly increased yield, decreased PD-L1, decreased IL-10) stand in sharp contrast to the lack of in vivo efficacy -- something the authors must address more strongly in the discussion.

Answer) According to your remarks, we have discussed this issue in lines 429–438.

It is also confusing that 7 patients were enrolled but results are only displayed for 3 patients. Are the authors going to display results for the other 4 patients, or at least of all 4 HLA-A*24:02. 3/4 of these patients were tetramer positive, but only 1/4 were IFN-y positive.

Answer) We have discussed this issue in lines 184–185, 194–195, and 196–197 of the revised manuscript. The results of WT1 tetramer and ELISpot assays have been added as Figure 6 and described in lines 334–361.

In terms of citations, many of the dendritic cell cancer therapies cited are self-referential, and some thought should be put into citing other high profile advances in the field, such as https://stm.sciencemag.org/content/10/436/eaao5931?et_cid=1967027&et_rid=314188761.

Answer) We have cited additional studies in lines 59–61 and 64–66.

Round 2

Reviewer 2 Report

The authors have modified and improved the manuscript accordingly. Therefore, I recommend its publication in this journal. Thanks.

Author Response

We thank you for your review.